# A Review on Pathogens and Necessary Diagnostic Work for Bleb-Related Infections (BRIs)

**DOI:** 10.3390/diagnostics12092075

**Published:** 2022-08-27

**Authors:** Stylianos A. Kandarakis, Leonidas Doumazos, Dimitra Mitsopoulou, Mario A. Economou, Ioanna Mylona, Chrysostomos Dimitriou, Petros Petrou, Ilias Georgalas

**Affiliations:** 1First Department of Ophthalmology, G. Gennimatas Hospital, National and Kapodistrian University of Athens, 115 27 Athens, Greece; 2Sophiehemet Hospital, 114 86 Stockholm, Sweden; 3Department of Clinical Neuroscience, Division of Ophthalmology and Vision, Karolinska Institutet, 171 77 Stockholm, Sweden; 4Department of Ophthalmology, General Hospital of Katerini, 601 00 Katerini, Greece; 5Ophthalmology Department, Colchester Eye Centre of Excellence, East Suffolk and North Essex NHS Foundation Trust, Turner Road, Mile End, Colchester CO4 5JR, UK

**Keywords:** bleb related infection, glaucoma filtration surgery, blebitis, trabeculectomy complications

## Abstract

At the present time, as newer techniques and minimally invasive procedures gain popularity among anterior segment surgeons for regulating intraocular pressure, trabeculectomy still has a leading role in glaucoma surgery. Trabeculectomy retains a highly successful and safe profile; however, one of the major complications includes bleb-related infections (BRIs). To date, the most common pathogens remain *Gram-positive cocci*, but the list of pathogens that have been identified in the literature includes more than 100 microorganisms. Because antibiotic use is more widespread than ever before and our ability to identify pathogens has improved, the pathogen spectrum will broaden in the future and more pathogens causing BRIs will be described as atypical presentations. The scope of this review was to identify all pathogens that have been described to cause bleb-related infections to date, as well as focus on the risk factors, clinical presentation, and various available diagnostic tools used for an appropriate diagnostic workup.

## 1. Introduction

Glaucoma filtration surgery (GFS) remains one of the major procedures within the glaucoma surgeons armamentarium to surgically reduce intraocular pressure (IOP), restrain the progression of glaucomatous optic neuropathy, and prevent further visual field loss [1]. While newer techniques and minimally invasive procedures have gained popularity among anterior segment surgeons, trabeculectomy still has a leading role in surgical intervention and retains a highly successful and safe profile [1,2]. One of the major complications following GFS is bleb-related infections (BRI), which may compromise surgical success and increase the risk of poor post-surgical vision [3]. The clinical presentations of BRIs include blebitis (stage I), as was first introduced by Brown et al. in 1994, and bleb-associated endophthalmitis (BAE), with anterior chamber (stage II) and vitreous (stage III) involvement accompanying the latter form of the infection [4]. Manifestations of the infection involve the local mucopurulent bleb infiltration with or without extension to intraocular inflammation, dependent on staging. BAE is further classified as early- or late-onset, with a cut-off point corresponding to one month postoperatively [4,5]. The filtering bleb is purportedly recognized as the portal entry of causative pathogens such as *Staphylococcus aureus*, *Staphylococcus epidermidis*, *Streptococcus species*, and *Haemophilus influenza*, which are the most common pathogens associated with BRIs, but in recent years, there have been multiple case reports describing rarer pathogens [6,7,8,9,10,11]. The goal of this review was to identify and categorize all of the pathogens reported in the literature to date, focus on the sampling procedures that have been described, and highlight the clinical characteristics of BRIs.

## 2. Classification of BRIs

BRIs are clinically categorized into localized inflammation with various degrees of anterior chamber involvement, namely blebitis and bleb-associated endophthalmitis (BAE), in which there is vitreous involvement [12]. A more detailed classification has been suggested based on the extent of the infection. Stage I, when localized inflammation of the bleb is present, is associated with a milky-white appearance of the bleb, mucopurulent discharge, and erythema around the bleb without anterior chamber or vitreous involvement [13,14]. In stage II, there is involvement of the anterior chamber with cells, flare or even hypopyon and in stage III, there is involvement of the vitreous with cellular infiltration [4,13,14]. Further subdivisions of stage III have been proposed, IIIa and IIIb, with mild and marked vitreous involvement, respectively [3]. Clinically, in stage IIIa the fundus is visible and no opacity is detectable in B-mode ultrasonography, while in IIIb, the vitreous is too opaque with the detection of opacity in ultrasonography [3,15,16].

Based on the postsurgical timeframe of BAE, it is crucial to consider that it may appear even years after GFS and, for that reason, it is further classified as early- or late-onset with a cut-off point corresponding to one month postoperatively [17]. This distinction is important, as early onset BAE has been shown to be caused by less virulent pathogens, similar to those from cataract surgeries, such as staphylococcus epidermidis [9,18,19]. In contrast, late-onset BAE is provoked by more virulent pathogens, such as *streptococcus species* and gram-negative bacteria such as *Hemophilus influenzae*. Therefore, late-onset BAE tends to have a poorer prognosis compared to endophthalmitis caused by other surgeries [4,18,20].

## 3. Risk Factors for BRIs

BRI can occur years after a GFS, even more than four decades after surgery [21], rendering it a lifetime concern [22]. Hence, it is important that all possible risk factors are minimized. Most notable risk factors include the following.

## 4. Location of the Filtering Bleb

The inferior and nasal location of bleb creation are associated with the highest risk for BRI. This is due to its exposure to the lacrimal lake and its normal flora and the increased subjection to mechanical stress from the lower lid [23,24]. Superiorly located blebs are protected from the superior lid and have a much lower incidence of BRI than those that are inferiorly located (1.3% vs. 7.8% per patient-year) [25]. Based on current data, inferiorly located blebs are considered high-risk procedures for BRIs and are rarely applied worldwide.

## 5. Conjunctival Approach

There are two approaches regarding bleb creation in trabeculectomy, namely fornix-based and limbus-based flaps. It has been clearly reported that limbus-based flaps have a much higher risk for infections [5]; a comparative study between the two techniques by Kuroda et al. showed 8% (limbus) vs. 0% (fornix) incidence of infection [26], while a similar study comparing the outcomes of glaucoma filtration surgery with different approaches reported a decrease in the incidence of BRI after changing the surgical technique from limbus-based to fornix-based (5.7% to 1.2%) [27].

## 6. Thin and Leaking Bleb

In the event of a leaking bleb, the barrier between the pathogen rich tear-film and the interior of the eye is disrupted and pathogens can easily access the eye, causing infection [28]. A higher risk of BRI has been suggested when the bleb is paper-thin and avascular [29]. In a case-control study, comparing 55 cases of bleb-related infections with matched control eyes, the findings indicated that eyes with a BRI are 26 times more likely to have an accompanying leak than eyes without BRI, highlighting the strong link between a leaking bleb and the risk of contamination [30]. Regarding bleb leakage, it has been categorized into early- and late-onset leak [29]. Early bleb leaks tend to occur when the bleb is inferiorly located, when MMC has been applied and after suture lysis or needling procedures [31,32]. Comparing early bleb leakage with late-onset leakage, the latter is regarded as the most important risk factor for BRIs [5].

## 7. Antimetabolite Agents

Although the success rates of functioning filtering surgeries have increased with the use of antimetabolites (MMC and 5-FU), and increased rates of late-onset bleb leakage have also been reported, leading to a threefold greater risk of developing endophthalmitis [5]. It has been hypothesized that the altered innate immune barrier of the conjunctiva over the bleb is defective and prone to infection due to antimetabolites [33]. The risk ratio of BRI and late-onset leakage is estimated to 2.48 and 1.31 with MMC and 5-FU, respectively, based on a multicenter case control study that included 131 patients from 27 different surgeons [28]. Nevertheless, these risk rates have decreased due to the refining of surgical technique in recent years [27].

## 8. Other Risk Factors

Conjunctivitis, blepharitis, prior history of recurrent bleb infection, and chronic use of topical antibiotics are all strongly associated with increased risk for BRI [28,29,34]. Another interesting finding when assessing risk factors for BRI is the role of phacoemulsification when combined with glaucoma filtration surgery. Trabeculectomy as a standalone procedure was found to be less protective for BRI while; when combined with phacoemulsification, the blebs were thicker, and thus less prone to leakage and had lower incidence of infections [28]. Finally, it has been reported that myopic eyes with high axial lengths (average of 25.8 mm) show greater risk for BRI, probably due to their thinner sclera and conjunctiva, which allow for the pathogens to penetrate more easily [29].

## 9. Clinical Presentation

Since BRI can occur from months to years after GFS [35], patients may delay directly associating their symptoms with the previous surgery, and this may be detrimental to prognosis. Hence, patients must be informed beforehand to correlate symptoms such as redness, irritation, blurred vision and pain with their GFS. In case of BAE, these symptoms evolve within hours with the exacerbation of ocular pain, redness and loss of visual acuity [12]. Interestingly, brow ache, headache or external ocular inflammation, such as blepharitis, are reported within days or weeks in 35% of patients before the diagnosis of BRI or BAE and are considered prodromal clinical symptoms [36].

The clinical course of a BRI usually follows a progressive path, with early stages characterized by intense conjunctival hyperemia localized to the region of the filtering bleb. Later, the bleb loses its transparency and appears milky-white, filled with fibrino-purulent infiltrates. A Seidel test must be performed to detect any leakage. Inflammatory cells as well as hypopyon may be seen in the anterior chamber depending on the severity and the duration of the BRI, while inflammatory cells in the vitreous are considered endophthalmitis. Nevertheless, when hypopyon and an obvious bleb infection is seen, it is classified as endophthalmitis until proven otherwise [5,24,37].

## 10. Work-Up-Sampling

Evaluation of the clinical presentation is conducted via a slit-lamp biomicroscopy examination. This includes a thorough examination of the bleb area, performing a Seidel test to detect any bleb leakage, and of the anterior chamber and vitreous, during which any presence of inflammatory cells must be noted. Gonioscopy examination is useful for the detection of microhypopyon [24]. If a stage IIIb BRI has been classified (marked opacity of the vitreous due to inflammation on slit-lamp), an ultrasound B-scan of the posterior segment is indicated, as retrolental inflammation in suspected BAE is determined via choroidal thickening and ultrasound echoes in the vitreous [5,24].

## 11. Conjunctival Swabs

Microbiologic work-up is vital for the appropriate management of all BRI cases [16,38]. This consists of conjunctival swabs, anterior chamber and vitreous samples, with the latter ones being mandatory when the classification of BRI is II, IIIa or IIIb [15,16,38]. After detailed ocular examination, conjunctival samples are collected and sent for Gram, Giemsa staining and cultures. Pre-moistened with brain heart infusion, broth sterile swabs are used to collect bleb cultures, which then undergo microbiological analysis [4,39]. Cultures are made on blood agar (supplemented with 5% defibrinated sheep blood), chocolate agar, and Sabouraud dextrose agar, inoculated into thioglycolate medium and brain heart infusion broth. Swabs are also used to make smears for direct microscopic identification by Gram’s stain [39]. After daily examination, the inoculated blood agar, chocolate agar, thioglycolate broth, and brain heart infusion broth are incubated at 37 °C under 5% CO_2_ for 1–2 weeks before being discarded if no growth is detected [40]. Additionally, after 3 weeks, the inoculated sabouraud dextrose agar plates, which are incubated at 27 °C under biochemical oxygen demand, are discarded if there is no growth [39]. Cultures are considered positive in one of the following situations: (1) growth is confluent (more than 10 colonies) at the site of inoculation on solid media, (2) the organism is seen with direct microscopy in the smears, or (3) the same organism is detected on more than one medium [39,40].

After the isolates have been identified, an antibiotic susceptibility test should be performed to determine the most effective drug for therapy [40,41,42]. Apart from the minimum inhibitory concentration (MIC) of the antibacterial agents, the minimum bactericidal concentration should be determined as well. This is of great clinical importance, particularly in endophthalmitis, as concentrations should be 2–4 times higher than MIC in order to be effective [40].

It is important to compare the nature of the isolated organism and differentiate flora from the active invading pathogen. The ocular surface microbiota has a vertical stratification at the genus level, with opportunistic and environmental microorganisms near the surface, and potential pathogens (*Staphylococci*, *Cornyebacteriae*, and *Proteobacteria*) are accessible at a deeper level [43]; thus, a complete swabbing at different levels and not only over the affected bleb is necessary to obtain a precise characterization and differentiation from the active invading pathogen. Unfortunately, the swabbing site is rarely reported in related studies.

BRI cases with mixed infections cannot be excluded. However, mixed cases of endophthalmitis generally have a very low incidence, with just two cases reported out of a total of 330 cases in a recent study, both of them mixed infections of gram-negative bacteria and fungi [44].

Although a full microbiologic work-up is recommended for the optimal management of BRI, in clinical practice, conjunctival swabs are not always performed. A survey conducted by members of the American Glaucoma Society on 2001 showed that only 28% of the clinicians always took conjunctival samples in blebitis, whereas 34% almost never or never took any conjunctival cultures. [45] Similarly, a survey performed in the United Kingdom in 2012 showed that 26% of clinicians did not acquire conjunctival samples [46]. Both surveys reflect the current clinical practice among clinicians in both the United States and the United Kingdom and verify that determining specific antibiotic treatment based on conjunctival swabs is not considered as useful or trustworthy due to the high chances of contamination. However, if treatment with antibiotics is commenced, then the chances for a future successful culture are considerably reduced; hence, the proper timing for the sampling procedure should always be adhered to since there is a chance that blind treatment with a wide-spectrum antibiotic may not be sufficient and could potentially lead to widespread bacterial resistance to these agents.

## 12. Aqueous and Vitreous Samples

Aqueous or vitreous samples are almost never taken when BRIs do not show signs of intraocular extension and are limited to the bleb area (blebitis, stage I) [45]. Conjunctival samples in cases that are classified as stage II or III are rarely collected according to guidelines, as it has been shown that the culture results from surface samples compared to those from intraocular samples (aqueous and vitreous) have significant discrepancies, ranging from 26% to 0% [33,47]. In general, microbiologic work-up is a time-consuming process and has poor sensitivity for bacterial detection, with negative culture results ranging from 21% to 86%, rendering the management of infective endophthalmitis challenging [5,48].

Apart from traditional microbiologic assays, molecular diagnostic tools are also available for the detection of culture negative cases via real-time polymerase chain reaction (PCR) [42,49,50]. Bispo et al., in a study with 31 samples, were able to estimate an improvement from 47.6% to 95.3% in detecting bacteria using real-time PCR [49]. Moreover, Cornut et al., in a comparative study with 7 patients with BRI, were able to obtain positive results and identify pathogens in all 7 cases using bacteria PCR technique, whereas traditional microbiologic culture revealed negative cultures in 6 out of 7 cases [51]. Pan-bacterial PCR has been used as a complementary technique to traditional microbiologic culture, providing a concomitant increase in detection sensitivity of 21% [6]. Additionally, for the diagnosis of fungal endophthalmitis, pan-fungal PCR using internal transcribed spacer (ITS) primers is considered a highly sensitive method, as fungi are known to have low sensitivity yields in conventional methods [40,52].

The advantages of PCR in aqueous and vitreous samples include the delivery of fast results (within 90 min), as well as a much higher specificity and sensitivity compared to standard microbiologic culture [40,53,54]. These features make PCR a helpful tool, especially in cases where traditional cultures take weeks to provide results, such as in slow-growing bacteria such as *mycobacteria,* or in organisms that are difficult to culture, such as *Microsporidia*, *Propionibacterium acnes* and *Toxoplasma gondii*, in all of which the earlier initiation of treatment could provide a better visual outcomes [40,55].

Despite its advantages, PCR has some limitations that do not allow for it to be broadly used. These include an approximately ten times higher cost compared to that for traditional microbiologic cultures, as well as a possible delay of up to four days in the delivery of results using some techniques, such as pan-bacteria PCR [5]. Another limitation is the strong probability of generating false negative or false positive results; therefore, interpretation should be conducted cautiously, considering both the clinical evaluation and the microbiologic culture for the validation of the result [51,56]. Parameters affecting the results, resulting in false negatives, comprise the innate presence of PCR inhibitors in the sample, the difficulty of lysing bacteria and the detection threshold, whereas false positives may be produced from possible contamination at any stage, molecular biology reagents or handling [51]. However, if a strict methodology is followed, the false positive rate of pan-bacterial PCR can be reduced to a range between 0% and 2% [57]. Finally, PCR techniques are being developed to target the most virulent bacteria causing endophthalmitis (*Streptococcus pneumoniae* and *Staphylococcus Aureus*), as well as detecting the presence of antibiotic resistance genes such as methicillin-resistant Staphylococci by mecA PCR [51,58].

## 13. Pathogens

A significant number of different microorganisms have been isolated and described as a cause of BRIs. Pathogen prevalence and virulence are BRI-stage dependent; the more advanced and late-onset the BRI is, the more virulent the isolates may be [9]. Coagulase-negative Staphylococcus are mostly identified in early-onset BRIs, whereas late-onset BRIs are most commonly culture-positive for Streptococcus [5]. Late-onset BRE are causatively related not only to gram-positive cocci, but also to pathogens such as *Haemophilus* sp., *Serratia* sp., *Enterococcus* sp. [6,11,59,60]. In a 14-year retrospective consecutive case series conducted by Jacobs et al., positive cultures were correlated with poorer VA outcomes [18].

The culture-proven causative microorganisms of BRIs are presented in Table 1, Table 2 and Table 3, classified by Gram staining. It is important to mention that overall, in our search of the literature, cultures were not performed in 104 cases, while 433 cultures proved no isolate. The above finding highlights two main issues. Firstly, it reflects the trend in clinical practice of a significant number of physicians who, in many cases, do not request microbial cultures. Secondly, it shows the limitations of the cultures since a significant number of cases may turn out negative. Negative cultures should not delay treatment but appropriate management should be adjusted to clinical findings and the experience of the physician. Additionally, there is cumulative evidence that in culture-negative cases, Abiotrophia defectiva should be of concern, since a culture-positive outcome is twofold in vitreous cultures compared to these from aqueous sampling and when suspected, PCR testing should be acquired [36,61].

Gram-positive cocci, such as *Streptococcus* and *Staphylococcus* sp., and gram-negative microorganisms, including Moraxella and Haemophilus sp., were found to be the most common pathogens related to BRIs, as also stated in several studies [5,9,62,63,64] (Table 1).

Among *Streptococcus* sp., which represent around 385 isolates, *S. pneumoniae* and *S. viridans* share the top of the chain of command of the causative pathogens, while *S. mitis*, *S. sanguinis*, and *S. oralis* are isolated 25% less frequently. Kawakami et al. published a rare case of late-onset BRE due to *S. pseudopneumoniae*, diagnosed and distinguished from other streptococci by 16S rRNA sequencing [8]. In a retrospective consecutive case series study concerning late onset BRE by Leng et al., *Peptostreptococcus prevoti* was isolated with two other coagulase-negative *Staphylococcus* sp. [9].

Regarding Staphylococcal BRIs, *S. aureus* is the evident isolate in almost half of the cases, followed by *S. epidermidis*. Studies with culture-proven coagulase-negative or -positive Staphylococcus are described in Table 1 within the *“NS, other, unidentifiable” staphylococcal* sp., which are equal to *S. aureus* cases. Not further specified gram-negative bacillus and species, and gram-positive bacillus and cocci are noted as “*NS cocci & rods, other* sp.”. Nocardia BRI cases are as frequent as those of *Gemella* sp. Interestingly, Ifantides et al. were the first to describe a case of Nocardia exalbida and Nocardia abscessus complex, as revealed from six cultures, in a patient with a history of trabeculectomy and bleb revision, who experienced blebitis [7].

In regards to gram-negative isolates (categorized in Table 2), *Moraxella* and *Haemophilus* sp. predominated in the majority of the culture-positive cases, where *Pseudomonas* and *Serratia* sp. ensued with 24 and 16 cases, respectively. Both the latter mentioned microorganisms are related to poorer VA outcomes compared to BRE cases caused by *Moraxella* sp., an outcome that is also evident in Pseudomonas BRE cases [18,65,66]. The preponderance of *Moraxella* sp. is apparent, designated for around 79 cases, while around 50 cultures did not further specify the species. *H. aegypticus*, *H. haemolyticus*, *H. pneumoniae* are the sole cases of the total 63 culture—-proven cases related to *Haemophilus* sp., as the majority are represented by *H. influenzae*. Twenty-eight cases of gram-negative BRI cases are described as “*NS cocci & rods, other* sp.” in Table 2.

Rare and uncommon microorganisms, causatively related to both blebitis and endophthalmitis, have also been reported in some case reports and case series. In a recent case report, Yang et al. referred to a patient with late-onset stage II blebitis with concomitant bleb perforation, ascribed to Capnocytophaga canimorsuss, an anaerobic gram-negative bacillus, transmitted by his canine saliva [11]. Streptococcus mitis presents with a moderate frequency concerning BRIs and we have reported 17 culture-proven cases. Among them, a late-onset BRE attributed to Streptococcus mitis, which possibly translocated from respiratory flora due to continuous positive airway pressure (CPAP), was described by Berg et al. [67]. Furthermore, skin flora microorganisms of low virulence, such as Dermabacter hominis and Kocuria rosea, have also been identified in BRE cases [6]. Moreover, as the role of rRNA sequencing in BRI diagnostic algorithm is gaining importance, more rare pathogens involved in BRIs will be revealed, as in the case of a late-onset BRE caused by Rothia mucilaginosa, a gram-positive coccus [10]. Finally, fungi and *mycobacteria* sp. are recognized as a scarce cause of BRIs and their management remains a great challenge [35,36,55,68,69,70,71,72,73]. These reported cases are summarized in Table 3.

## 14. Conclusions

BRIs remain one of the most serious complications after glaucoma filtration surgery, are considered a vision-threatening situation, and may severely compromise surgical success. Prognosis varies and depends on multiple factors but undoubtedly, early detection remains a crucial factor on the final outcome [5,36,63]. To ensure the rapid recognition of an evolving infection, both patients and physicians should be vigilant for the signs that characterize early stages of BRIs. Informing the patients, who have undergone glaucoma filtration surgery, that they carry a lifelong risk of infection at the site of surgery is critical, as well as advising them to seek urgent consultation when first signs appear [21,45]. Physicians should recognize the clinical presentation of a BRI, identify the stage, perform clinical work-up, and start appropriate treatment as soon as possible [45,46]. Being familiar with the risk factors associated with the BRIs may prove helpful in these situations. The specific characteristics (vascularity, thickness, integrity), the location of the bleb (inferior versus anterior) and the use of antimetabolites (agent type, duration, concentration) remain among the most important risk factors that should be identified by general ophthalmologists [29]. The post-surgical time period between surgery and the beginning of the BRI is also significant, since late onset BRIs involve different pathogens and have a worse prognosis than earlier infections. In general, when considering culture-positive pathogens for BRIs, gram-positive cocci should be the main focus. While more than one hundred different pathogens have been described in the literature and are included in our review, gram-positive cocci remain by far the most commonly reported microorganisms. Pathogens involved in the early stages of BRI are usually less virulent and predominately belong to the coagulase-negative Staphylococcus species, while pathogens in late-onset BRIs belong mainly to Streptococcus sp., followed by *Haemophilus* sp., *Serratia* sp., and *Enterococcus* sp. [5]. Because antibiotic use is more widespread than ever before and our ability to identify pathogens has improved, we believe that the pathogen spectrum will broaden in the future and more pathogens causing BRIs will be described as atypical presentations. For this reason, it is important to use efficiently all available diagnostic tools, including microbiological cultures and the PCR of aqueous and vitreous when appropriate. Equally, physicians should consider the limitations and restrictions of various forms of diagnostic testing, including the level of specificity and sensitivity, as well as cost-effectiveness. Fortunately, new directions are being offered with the advent of new methods, such as, for example, those that fall in the domain of metageniomics [74]: a new methodology of high-throughput DNA sequencing, which provides taxonomic and functional profiles of microbial communities without a cell culture in the laboratory. These new methods have been employed for delicate examinations of extreme importance, such as detecting the presence of microorganisms in the storage media of human donor corneas, and have shown superior results [75].

A limitation of this study is that the search strategy did not include clinical trials, although relevant valid results would have been published in peer-reviewed journals, so the decision was made not to expand the search since there was a risk of including misleading results.

## 15. The Literature Search

We concluded a Pubmed search from 1961 to 2022, and a Scopus review of the literature from 1970 to 2022. The terms used covered a wide spectrum of terms associated with bleb-related infections, including: blebitis, bleb associated infections (BAE), complications of glaucoma filtration surgery, endophthalmitis, bleb related infections (BRIs), keratitis, scleritis, blepharitis, and glaucoma. Reference lists from the recovered articles were also used to identify cases that may have not been included in our initial search.

## Figures and Tables

**Table 1 diagnostics-12-02075-t001:** Grampositive pathogens causing BRIs.

**Gram Positive Pathogens**	**Study Represented by First Author, Year of Publication, Number of Eyes in Parenthesis**
*Abiotrophia defectiva*	Lee et al, 2015 (1); Mustafi et al, 2021 (1)
***Bacillus* sp.**	
*Bacillus cereus*	Miller et al, 2008 (1)
other	Moloney et al, 2014 (1)
***Corynebacterium* sp.**	
*Corynebacterium macginleyi*	Del Barrio et al, 2019 (1); Qin et al, 2018 (1)
*Corynebacterium xerosis*	Ramakrishnan et al, 2011 (3)
other	Benz et al, 2004 (2); Leng et al, 2011 (2); Sagara et al, 2016 (7); Sagara et al, 2017 (5); Yamamoto et al, 2013a (7); Yamamoto et al, 2013b (7)
*Dermabacter hominis*	Brillat-Zaratzian et al, 2013 (1)
*Diphtheroids*	Solomon et al, 1999 (1)
***Enterococcus* sp.**	
*Enterococcus/Streptococcus faecalis*	Brillat-Zaratzian et al, 2013 (3); Busbee et al, 2004 (1); Chen et al, 2021 (3); Greenfield et al, 1996 (1); Kangas et al, 1997 (2); Mandelbaum et al, 1985 (4); Ohtomo et al, 2015 (1); Tang et al, 2007 (1) VRE
other, NS	Benz et al, 2004 (7); Busbee et al, 2004 (4); Ciulla et al, 1997 (2); Jacobs et al, 2011 (6); Kwon et al, 2018 (3); Leng et al, 2011 (5); Matsuo et al, 2002 (1);
	Sagara et al, 2016 (4); Sagara et al, 2017 (3); Song et al, 2002 (9); Yamamoto et al, 2013a (4); Yamamoto et al, 2013b (4)
***Gemella* sp. (5)**	
*Gemella haemolysans*	Sagara et al, 2016 (1); Sagara et al, 2017 (1); Sawada et al, 2009 (1); Yamamoto et al, 2013a (1)
*Gemella morbillorum*	Sawada et al, 2009 (1)
*Kocuria rosea*	Brillat-Zaratzian et al, 2013 (1)
*Micrococcus luteus*	Sagara et al, 2016 (1); Sagara et al, 2017 (1); Yamamoto et al, 2013a (1)
***Nocardia* sp. (7)**	
*Nocardia abscessus*	Ifantides et al, 2015 (1)
*Nocardia exalbida*	Ifantides et al, 2015 (1)
other, NS	William et al, 2017 (1)
*Peptostreptococcus prevoti*	Leng et al, 2011 (1)
***Propionibacterium* sp. (19)**	
*Propionibacterium acnes*	Al-Turki et al, 2010 (6); Benz et al, 2004 (1); Brillat-Zaratzian et al, 2013 (1), Busbee et al, 2004 (1); Ciulla et al, 1997 (3); Higginbotham et al, 1996 (1); Kangas et al, 1997 (1); Leng et al, 2011 (1); Poulsen et al, 2000 (2); Wallin et al, 2013 (1)
other, NS	Kwon et al, 2018 (1)
***Staphylococcus* sp. (296)**	
*Staphylococcus aureus* (122)	Benz et al, 2004 (1); Brillat-Zaratzian et al, 2013 (1), Busbee et al, 2004 (2); Ciulla et al, 1997 (7); Gedde et al, 2001 (1); Kerr et al, 2018 (1); Kuang et al, 2008 (2); Lehmann et al, 2000 (1); Leng et al, 2011 (8); Luebke et al, 2018 (3);
	Mandelbaum et al, 1985 (2); Matsuo et al, 2002 (1); Pierre et al, 2010 (1) MRSA; Poulsen et al, 2000 (2); Ramakrishnan et al, 2011 (6); Sagara et al, 2016 (12) MRSA incl; Sagara et al, 2017 (8) MRSA incl; Sharan et al, 2009 (1);
	Song et al, 2002 (4); Waheed et al, 1997 (1); Waheed et al, 1998 (12); Wallin et al, 2013 (5); Wolner et al, 1991 (8); Yamamoto et al, 2013a (18) MRSA incl; Yamamoto et al, 2013b (14) MRSA incl
*Staphylococcus epidermidis* (57)	Al-Turki et al, 2010 (7); Benz et al, 2004 (8); Busbee et al, 2004 (8); Ciulla et al, 1997 (11); Greenfield et al, 1996 (1); Kangas et al, 1997 (7); Luebke et al, 2018 (1); Ohtomo et al, 2015 (1); Ramakrishnan et al, 2011 (2); Solomon et al, 1999 (2)
NS, other, unidentifiable (117)	Benz et al, 2004 (4); Busbee et al, 2004 (1); Gupta et al, 2014 (1); Higginbotham et al, 1996 (1); Jacobs et al, 2011 (9); Kuang et al, 2008 (1); Kwon et al, 2018 (7); Leng et al, 2011 (12); Poulsen et al, 2000 (1); Sagara et al, 2016 (14) MRSE incl;
	Sagara et al, 2017 (7) MRSE incl; Solomon et al, 1999 (1); Song et al, 2002 (6); Waheed et al, 1997 (2); Wallin et al, 2013 (8); Yamamoto et al, 2013a (14) MRSE incl; Yamamoto et al, 2013b (14) MRSE incl; Yap et al, 2016 (4); Waheed et al, 1998 (10)
***Streptococcus* sp. (385)**	
*a-hemolytic streptococcus*	Lehmann et al, 2000 (1); Poulsen et al, 2000 (1)
*b-hemolytic streptococcus*	Kuriyan et al, 2014 (2); Mandelbaum et al, 1985 (2)
*Streptococcus albus*	Lehmann et al, 2000 (1)
*Streptococcus agalactiae*	Al-Turki et al, 2010 (1); Leng et al, 2011 (1); Song et al, 2002 (4)
*Streptococcus caprae*	Ramakrishnan et al, 2011 (2)
*Streptococcus hominis*	Kangas et al, 1997 (1); Luebke et al, 2018 (1); Ramakrishnan et al, 2011 (3)
*Streptococcus intermedius*	Leng et al, 2011 (1); Mochizuki et al, 2009 (1); Ramakrishnan et al, 2011 (1)
*Streptococcus mitis* (17)	Berg et al, 2018 (1), Brillat-Zaratzian et al, 2013 (1); Busbee et al, 2004 (4); Kangas et al, 1997 (1); Leng et al, 2011 (4); Ohtomo et al, 2015 (2); Song et al, 2002 (4)
*Streptococcus mutans*	Busbee et al, 2004 (1)
*Streptococcus oralis* (11)	Brillat-Zaratzian et al, 2013 (1); Leng et al, 2011 (5); Song et al, 2002 (5)
*Streptococcus pneumoniae* (59)	Al-Turki et al, 2010 (11); Beck et al, 2000 (1); Benz et al, 2004 (3); Brillat-Zaratzian et al, 2013 (2), Busbee et al, 2004 (6); Ciulla et al, 1997 (2); Gedde et al, 2001 (1); Kangas et al, 1997 (3); Kuriyan et al, 2014 (4); Lehmann et al, 2000 (2);
	Leng et al, 2011 (1); Luebke et al, 2018 (1); Mandelbaum et al, 1985 (4); Miller et al, 2004 (6); Moloney et al, 2014 (3); Ohtomo et al, 2015 (2); Ramakrishnan et al, 2011 (3); Song et al, 2002 (1) Waheed et al, 1998 (3)
*Streptococcus pseudopneumoniae*	Kawakami et al, 2013 (1)
*Streptococcus pyogenes*	Al-Turki et al, 2010 (1); Waheed et al, 1998 (1)
*Streptococcus saccharolyticus*	Ramakrishnan et al, 2011 (1)
*Streptococcus sanguinis* (15)	Brillat-Zaratzian et al, 2013 (1); Busbee et al, 2004 (5); Greenfield et al, 1996 (3); Higginbotham et al, 1996 (2); Kangas et al, 1997 (2); Leng et al, 2011 (1); Song et al, 2002 (1)
*Streptococcus sciuri*	Al-Turki et al, 2010 (1)
*Streptococcus* sp. group A	Ohtomo et al, 2015 (1)
*Streptococcus* sp. group B	Busbee et al, 2004 (1)
*Streptococcus* sp. group C	Beck et al, 2000 (2)
*Streptococcus* sp. group G	Beck et al, 2000 (1); Kangas et al, 1997 (1); Leng et al, 2011 (2); Sharan et al, 2009 (1); Song et al, 2002 (1)
*Streptococcus viridans* group (VGS) (64)	Al-Turki et al, 2010 (10); Benz et al, 2004 (10); Lehmann et al, 2000 (1); Kangas et al, 1997 (7); Kuang et al, 2008 (1); Kuriyan et al, 2014 (11);Leng et al, 2011 (3); Mandelbaum et al, 1985 (3); Poulsen et al, 2000 (1);
	Ramakrishnan et al, 2011 (1); Solomon et al, 1999 (1) non hemolytic; Song et al, 2002 (2); Waheed et al, 1998 (4)
NS, other, unidentifiable (171)	Benz et al, 2004 (2); Ciulla et al, 1997 (10); Jacobs et al, 2011 (21); Kwon et al, 2018 (22); Leng et al, 2011 (1); Mandelbaum et al, 1985 (3); Ohtomo et al, 2015 (1); Poulsen et al, 2000 (1); Sagara et al, 2016 (25);
	Sagara et al, 2017 (18); Song et al, 2002 (3); Wallin et al, 2013 (9); Yamamoto et al, 2013a (25); Yamamoto et al, 2013b (24); Yap et al, 2016 (6)
*NS cocci & rods, other* sp.	Dimacali et al, 2020 (2); Sagara et al, 2016 (2); Sagara et al, 2017 (1); Waheed et al, 1997 (1); Yamamoto et al, 2013a (2)

**Table 2 diagnostics-12-02075-t002:** Gram-negative pathogens causing BRIs.

**Gram Negative Pathogen**	**Study Represented by First Author, Year of Publication, Number of Eyes in Parenthesis**
***Acinetobacter* sp. (3)**	
*Acinetobacter calcoaceticus*	Gopal et al, 2003 (1)
other, NS	Waheed et al, 1998 (2)
*Alcaligenes faecalis*	Solomon et al, 1999 (1)
*Bacteroides* spp.	Parker et al, 2019 (1)
Capnocytophaga canimorsus	Yang et al, 2021 (1)
***Enterobacter* sp. (8)**	
*Enterobacter cloacae*	Ciulla et al, 1997 (2); Okhravi et al, 1998 (4)
other, NS	Leng et al, 2011 (1); Song et al, 2002 (1)
*Escherichia coli*	Solomon et al, 1999 (2)
***Haemophilus* sp. (63)**	
*Haemophilus aegypticus*	Busbee et al, 2004 (1)
*Haemophilus haemolyticus*	Wolner et al, 1991 (1)
*Haemophilus influenzae*	Al-Turki et al, 2010 (8); Ciulla et al, 1997 (2); Gedde et al, 2001 (1); Greenfield et al, 1996 (3); Lehmann et al, 2000 (1); Leng et al, 2011 (3); Kangas et al, 1997 (5); Kwon et al, 2018 (5); Mandelbaum et al, 1985 (7); Ramakrishnan et al, 2011 (1); Sagara et al, 2016 (4); Sagara et al, 2017 (2); Song et al, 2002 (1); Waheed et al, 1998 (2); Wolner et al, 1991 (1); Yamamoto et al, 2013a (4); Yamamoto et al, 2013b (4); Yap et al, 2016 (5)
*Haemophilus pneumoniae*	Busbee et al, 2004 (1)
other, NS	Moloney et al, 2014 (1)
***Klebsiella* sp. (4)**	
*Klebsiella pneumoniae*	Solomon et al, 1999 (2); Sridhar et al, 2014 (1)
other, NS	Kwon et al, 2018 (1)
***Moraxella* sp. (79)**	
*Moraxella catarrhalis*	Berrocal et al, 2001 (4); Ciulla et al, 1997 (6); Cornut et al, 2008 (2); Takahashi et al, 2019 (2); Valle et al, 2015 (1)
*Moraxella lacunata*	Cornut et al, 2008 (1); Waheed et al, 1998 (1)
*Moraxella liquefaciens*	Waheed et al, 1998 (1)
*Moraxella nonliquefaciens*	Barron et al, 2020 (1); Cornut et al, 2008 (2); Kerr et al, 2018 (1); Mandelbaum et al, 1985 (1);Takahashi et al, 2019 (1)
*Moraxella osloensis*	Berrocal et al, 2001 & 2002 (3)
other, NS (50)	Al-Turki et al, 2010 (2); Brillat-Zaratzian et al, 2013 (5); Busbee et al, 2004 (3); Cornut et al, 2008 (2); Higginbotham et al, 1996 (1); Jacobs et al, 2011 (8); Kwon et al, 2018 (11);Lehmann et al, 2000 (1); Leng et al, 2011 (7); Moloney et al, 2014 (2); Poulsen et al, 2000 (1); Solomon et al, 1999 (1); Song et al, 2002 (6)
*Morganella morganii*	Kuang et al, 2008 (1)
*Neisseria subflava*	Waheed et al, 1998 (1)
*Proteus mirabilis*	Leng et al, 2011 (1)
***Pseudomonas* sp. (24)**	
*Pseudomonas aeruginosa*	Bharathi et al, 2014 (1); Busbee et al, 2004 (3); Eifrig et al, 2003 (4); Gedde et al, 2001 (1); Greenfield et al, 1996 (1); Leng et al, 2011 (4); Mandelbaum et al, 1985 (2); Sagara et al, 2016 (1); Solomon et al, 1999 (1); Wolner et al, 1991 (1); Yamamoto et al, 2013a (1)
other, NS	Jacobs et al, 2011 (3); Poulsen et al, 2000 (1)
*Rhodococcus equi*	Al-Turki et al, 2010 (1)
*Rothia mucilaginosa*	Oie et al, 2016 (1)
***Serratia* sp. (16)**	
*Serratia marcescens*	Busbee et al, 2004 (2); Ciulla et al, 1997 (3); Leng et al, 2011 (3); Waheed et al, 1998 (1); Wolner et al, 1991 (1)
other, NS	Jacobs et al, 2011 (3); Song et al, 2002 (3)
*Fastidious gram-negative rods*	Benz et al, 2004 (11)
*NS cocci & rods, other* sp	Benz et al, 2004 (2); Jacobs et al, 2011 (15); Lehmann et al, 2000 (2); Sagara et al, 2016 (1); Sagara et al, 2017 (1); Song et al, 2002 (1); Wallin et al, 2013 (5); Yamamoto et al, 2013a (1)

**Table 3 diagnostics-12-02075-t003:** Rare and uncommon pathogens associated with BRIs.

**Pathogen**	**Study Represented by First Author, Year of Publication, Number of Eyes in Parenthesis**
*Aspergillus niger*	Dimacali et al, 2020 (1)
***Candida* sp.**	
*Candida albicans*	Benz et al, 2004 (2); Busbee et al, 2004 (1)
other, NS	Ciulla et al, 1997 (1)
*Fusarium* sp.	Mandelbaum et al, 1985 (1); Poulsen et al, 2000 (1)
*Lecythophora mutabilis*	Scott et al, 2004 (1)
***Mycobacteria* sp.**	
*Mycobacterium chelonae*	Gedde et al, 2001 (1)
*Mycobacterium tuberculosis*	Seth et al, 2020 (1)
other, NS	Benz et al, 2004 (1)
*Anaerobic bacteria* (unidentified)	Sagara et al, 2016 (1); Sagara et al, 2017 (1); Yamamoto et al, 2013a (1)
Others	Dimacali et al, 2020 (2); Yamamoto et al, 2013b (6), Yap et al, 2016 (10)

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
