# Peer review of "A Review on Pathogens and Necessary Diagnostic Work for Bleb-Related Infections (BRIs)"

_diagnostics, 2022, doi:10.3390/diagnostics12092075_

Round 1

Reviewer 1 Report

11: conjunctival swaps

“swaps” typo error “swabs”

Please highlight the importance of the timing of the sampling procedure in relation to the antibiotic treatment (given that antibiotic treatment increases the chances of a negative culture).

Please mention the recommended site for sampling of the conjunctiva (i.e. over the affected bleb, from the inferior fornix, from neighbouring conjunctiva, etc) to compare the nature of the organism isolated and differentiate flora from the active invading pathogen.

12. Aqueous and Vitreous Samples

Please mention the degree of correlation in the published studies between the isolated pathogens from conjunctival swabs and pathogens isolated from the AC or vitreous. This may highlight the importance of conjunctival swabs in the diagnosis and management of BRI.

Please comment on the possibility of mixed infection (polymicrobial infection) in BRI and the possible pathogens involved.

Author Response

Response to reviewer comments

We wish to thank the anonymous reviewers for their time and careful review of our manuscript. Changes in the manuscript have been highlighted in yellow. A detailed response to each and every comment follows:

Reviewer #1

  1. >11: conjunctival swaps

“swaps” typo error “swabs”

Response: Thank you for your comment, this has been corrected

  1. Please highlight the importance of the timing of the sampling procedure in relation to the antibiotic treatment (given that antibiotic treatment increases the chances of a negative culture).

Response: Thank you for your comment, your suggestion has been included in the revised manuscript (lines 186-190)

  1. Please mention the recommended site for sampling of the conjunctiva (i.e. over the affected bleb, from the inferior fornix, from neighbouring conjunctiva, etc) to compare the nature of the organism isolated and differentiate flora from the active invading pathogen.

Response:  Thank you for your constructive suggestion, this recommendation has been included in the revised manuscript (lines 177-185)

  1. Aqueous and Vitreous Samples

Please mention the degree of correlation in the published studies between the isolated pathogens from conjunctival swabs and pathogens isolated from the AC or vitreous. This may highlight the importance of conjunctival swabs in the diagnosis and management of BRI.

Response: Thank you for your comment this information has been inserted in the revised manuscript (lines 201-205)

  1. Please comment on the possibility of mixed infection (polymicrobial infection) in BRI and the possible pathogens involved.

Response: Thank you for your suggestion, this has been added in the revised manuscript (lines 185-188)

Reviewer 2 Report

Authors wrote an interesting article.

Few improvements are needed.

BRIs are clinically categorized into localized inflammation (or infections?) with various degree of anterior chamber involvement namely blebitis and bleb associated endophthalmitis (BAE) in which there is vitreous involvemen

Please improve the English level

Please improve the limitations of the study

Please add sentence or section about future way do improve diagnostics in infection. I'd suggest to add a sentence regarding metageniomics. Here few papers. PMID: 31276030 and  PMID: 31179394

Author Response

Response to reviewer comments

We wish to thank the anonymous reviewers for their time and careful review of our manuscript. Changes in the manuscript have been highlighted in yellow. A detailed response to each and every comment follows:

Reviewer #2

Few improvements are needed.

  1. >Please improve the English level

Response: Thank you for your suggestion, changes have been made throughout the manuscript with the aid of a native speaker, too numerous to mention

  1. >Please improve the limitations of the study

Response: Thank you for your helpful comment, a limitations section has been included in the revised article. (lines 360-363)

  1. >Please add sentence or section about future way do improve diagnostics in infection. I'd suggest to add a sentence regarding metageniomics. Here few papers. PMID: 31276030 and PMID: 31179394

Response: Thank you for your helpful comment, metageniomics have been referenced with your two suggested papers

Round 2

Reviewer 2 Report

Approved